# Exploring the Complex Relationship between Diabetes and Cardiovascular Complications: Understanding Diabetic Cardiomyopathy and Promising Therapies

**DOI:** 10.3390/biomedicines11041126

**Published:** 2023-04-07

**Authors:** Nilanjan Ghosh, Leena Chacko, Hiranmoy Bhattacharya, Jayalakshmi Vallamkondu, Sagnik Nag, Abhijit Dey, Tanushree Karmakar, P. Hemachandra Reddy, Ramesh Kandimalla, Saikat Dewanjee

**Affiliations:** 1Molecular Pharmacology Research Laboratory, Department of Pharmaceutical Technology, Jadavpur University, Kolkata 700032, India; 2BioAnalytical Lab, Meso Scale Discovery, Rockville, MD 20850-3173, USA; 3Advanced Pharmacognosy Research Laboratory, Department of Pharmaceutical Technology, Jadavpur University, Kolkata 700032, India; 4Department of Physics, National Institute of Technology, Warangal 506004, India; 5Department of Biotechnology, Vellore Institute of Technology (VIT), School of Biosciences & Technology, Tiruvalam Road, Vellore 632014, India; 6Department of Life Sciences, Presidency University, Kolkata 700073, India; 7Dr. B C Roy College of Pharmacy and Allied Health Sciences, Durgapur 713206, India; 8Texas Tech University Health Sciences Center, Lubbock, TX 79430-0002, USA; 9Department of Biochemistry, Kakatiya Medical College, Warangal 506007, India

**Keywords:** diabetes mellitus, diabetic cardiomyopathy, heart failure, hyperglycemia, insulin resistance, lipotoxicity

## Abstract

Diabetes mellitus (DM) and cardiovascular complications are two unmet medical emergencies that can occur together. The rising incidence of heart failure in diabetic populations, in addition to apparent coronary heart disease, ischemia, and hypertension-related complications, has created a more challenging situation. Diabetes, as a predominant cardio-renal metabolic syndrome, is related to severe vascular risk factors, and it underlies various complex pathophysiological pathways at the metabolic and molecular level that progress and converge toward the development of diabetic cardiomyopathy (DCM). DCM involves several downstream cascades that cause structural and functional alterations of the diabetic heart, such as diastolic dysfunction progressing into systolic dysfunction, cardiomyocyte hypertrophy, myocardial fibrosis, and subsequent heart failure over time. The effects of glucagon-like peptide-1 (GLP-1) analogues and sodium-glucose cotransporter-2 (SGLT-2) inhibitors on cardiovascular (CV) outcomes in diabetes have shown promising results, including improved contractile bioenergetics and significant cardiovascular benefits. The purpose of this article is to highlight the various pathophysiological, metabolic, and molecular pathways that contribute to the development of DCM and its significant effects on cardiac morphology and functioning. Additionally, this article will discuss the potential therapies that may be available in the future.

## 1. Introduction

Diabetes mellitus (DM) and its associated pathophysiological events have emerged as a leading risk factor for the development of diabetes cardiomyopathies (DCM) and subsequent heart failures, posing serious health concerns worldwide. The identification of a causal link between cardiomyopathy and diabetes within the cardio-renal metabolic syndrome has paved the way for a more promising and rational approach to the underlying pathophysiological pathways and treatment regimens. Retrospective studies indicate that a significant number of diabetic patients develop DCM due to vascular risk factors [1,2]. DCM is a complex pathological condition characterized by myocardial dysfunction in diabetic patients, in the absence of apparent coronary heart disease, valvular disease, dyslipidemia, and systemic hypertension. It is a distinct form of heart disease that is promoted by insulin resistance, compensatory hyperinsulinemia, and progressive hyperglycemia [3,4,5]. The pathology of DCM develops insidiously due to its silent progression in the initial stages, encompassing subtle structural and functional irregularities such as decreased diastolic compliance, cardiomyocyte hypertrophy, myocardial fibrosis, cardiomyocyte apoptosis, dysfunctional remodeling, and thickening of the capillary basement membrane. It includes a latent asymptomatic phase involving left ventricular (LV) hypertrophy, cardiac fibrosis, and a constellation of other characteristics mediated by myocardial collagen deposition, cardiac insulin signaling abnormalities, cardiac stiffness, mitochondrial dysfunction, endoplasmic reticulum stress (ERS), impaired calcium handling, abnormal coronary microcirculation, dysfunctional neurohumoral activation, inflammation, altered immune responses, diffused hypokinesis, an increase in free fatty acid (FFA) levels, advanced glycation end products (AGEs), and reactive oxygen species (ROS)accumulation [6,7] (Figure 1). DCM is associated with subclinical diastolic dysfunction that progresses to heart failure with normal ejection fraction and eventually evolves into systolic dysfunction accompanied by heart failure with reduced ejection fraction [8]. These significant metabolic and molecular changes make DCM a distinct entity apart from other severe cardiovascular syndromes. Furthermore, DCM can be classified into two major components: short-term physiological adaptation with metabolic alterations and degenerative changes of which the myocardium has limited capacity to repair.

The concept of DCM was first introduced by Shirley Rubler and colleagues in 1972 during the autopsy of four diabetic patients exhibiting symptoms of heart failure without any clear signs of known precipitating factors such as coronary artery or valve disease [3]. In 1974, a higher incidence of heart failure in diabetic women (5-fold) and men (2.4-fold) was reported, thereby establishing an initial epidemiological link between diabetes and heart failure [3,4]. The theory of prediabetes as a preceding phenomenon to type 2 DM (T2DM) has provided further insight into predicting DCM [9,10]. Prediabetes refers to a mild and transient hyperglycemic state that includes poor glucose tolerance. The clinical burden associated with several cardiovascular complications in T2DM patients has necessitated a focus on the joint management of T2DM and DCM. These rigorous theories of metabolic syndrome have spurred extensive exploration of molecular and pathophysiological mechanisms to explain the cluster of risk factors involved, presenting a challenge in the field. This review explores various molecular and metabolic pathways underlying cardiac dysfunctions in DCM and associated abnormalities or discrepancies caused by hyperglycemia, which may progress to heart failure. It also highlights several emerging potential pharmacotherapies that can be considered as conventional therapies face increasing limitations.

## 2. Epidemiology

Patients with diabetes are at a higher risk of developing cardiovascular diseases (CVD) compared to those without diabetes. It is worth mentioning both type 1 DM (T1DM) and T2DM could worsen diabetic hearts. However, heart failure is relatively rare in T1DM subjects as compared to T2DM subjects, which may be due to only T2DM representing hyperinsulinemia, explaining the difference in the clinical features of DCM found in patients affected by T2DM along with age factor. As estimated by the International Diabetes Federation in 2019, around 10% of the world’s population (537 million people) were diabetic, with 91% of them having type 2 diabetes (T2DM) [11]. However, it has been predicted that the number of diabetes cases will increase to 783 million by 2045, a staggering increase of 51% [11,12,13]. The prevalence of diabetes is similar in both high- and low-income countries. The risk of cardiovascular complications attributable to T2DM has increased from 5.4 to 8.7% between 1952 and 1998, as reported by the Framingham Heart Study in 2015 [13]. CVD was found to be the cause of death in 9.9% of T2DM patients, according to a report by Einarson and colleagues [14]. The prevalence of diastolic and myocardial dysfunction in T2DM patients has also risen to 40–60%, accounting for 85% of heart failure cases [15]. Adults with T2DM are two to three times more likely to develop cardiovascular complications than those without T2DM [16]. A report shows that 31% of individuals with DCM develop mortality or heart failure [17]. Thus, screening individuals with T2DM for cardiovascular risks could be an effective strategy for reducing mortality and CVD events in T2DM. Despite several therapeutic measures having been adopted to attenuate DCM in T2DM patients, the threat remains progressively unabated [7]. Preclinically effective molecules failed to be effective in the clinical milieu. Thus, there is an urgent need to discover/develop novel therapeutic strategies to attenuate DCM. In the subsequent sections of this manuscript, we have discussed the possibilities of novel therapeutic approaches to counteract DM.

## 3. Progression of DCM

Cardiomyopathy is a group of myocardial conditions characterized by mechanical and electrical dysfunction, with structural defects in cardiomyocytes that can result from genetic causes or improper ventricular hypertrophy, according to the American Heart Association [18]. The progression DCM involves three main stages: early, middle, and late. In the early stage, hypertrophic changes appear in the heart, along with diastolic dysfunction and normal ejection fraction. Cellular and metabolic changes occur without definite systolic dysfunction, initiated by hyperglycemia. These metabolic disturbances include elevated levels of FFAs, disturbed calcium homeostasis, depletion of glucose transporter (GLUT)-1 and GLUT-4, carnitine deficiency, and insulin resistance [19,20,21].

The middle stage is characterized mainly by myocardial hypertrophy and fibrosis due to collagen accumulation resulting from impaired collagen degradation. There is an increase in LV dimensions, wall thickness, and mass with diastolic dysfunction and moderately reduced ejection fraction. At the molecular and cellular level, this stage is accompanied by insulin resistance, formation of AGEs, elevated levels of the renin–angiotensin–aldosterone system (RAAS), and transforming growth factor-β1 (TGF-β1), along with reduced levels of insulin growth factor-1 (IGF-1), defects in calcium transport, and fatty acid metabolism. These alterations promote increased myocyte apoptosis, oxidative stress, maladaptive immune responses, mild cardiac autonomic neuropathy leading to loss of myocytes, and myocardial fibrosis leading to low ejection fraction. TGF-β1 plays a significant role in the upregulation of collagen expression, which produces significant LV remodeling characteristics [22,23]. The development of the late stage from the middle stage is usually characterized by an alteration in microvasculature compliance with various additional severities, converging several factors such as an increase in LV size, reduced cardiac performance, and coronary artery disease.

Hypertension, along with the onset of ischemic conditions, impairs both systolic and diastolic functions, ultimately leading to subsequent heart failure. The alterations in cardiac structure progress toward impaired coronary microcirculation accompanying recurrent microvascular spasm. Impaired myocardial insulin signaling is associated with reduced activation of endothelial nitric oxide synthase (NOS), which further exacerbates elevation in oxidative stress due to an increase in the levels of ROS and inflammation, accompanied by decreased NO levels. These conditions stimulate interstitial collagen deposition [24,25]. Sequential pathogenic events brought on by hyperglycemia in diabetic hearts are shown in Figure 2.

## 4. Cardiac Structural and Functional Anomalies in DCM

The development of DCM is marked by several structural abnormalities in the heart, which result from an increase in the size of heart muscle cells (cardiomyocyte hypertrophy) and enlargement of the LV [26]. LV concentric remodeling, which is characterized by the thickening of the LV wall, is the primary structural characteristic of DCM and is both a predictor and a preceding factor in the development of heart failure and other adverse cardiovascular events.

One of the early pathological abnormalities in DCM is diastolic dysfunction, which occurs when the heart is unable to relax and fill with blood properly [27,28]. This dysfunction is associated with diabetes-induced changes in cardiac lipid accumulation and calcium homeostasis, leading to ventricular stiffness and impaired relaxation. As diastolic dysfunction progresses to systolic dysfunction, the heart becomes unable to pump enough blood, resulting in reduced LV ejection fraction over time.

Systolic dysfunction, which is a later manifestation of DCM, is primarily dependent on myocyte loss. There is a consistent association between altered LV geometry, LV end-diastolic volume, and insulin resistance due to the effect of insulin on inducing hypertrophy and cellular growth, leading to interstitial fibrosis as a key factor in the pathogenesis of left ventricular hypertrophy (LVH) seen in advanced stages of DCM [29,30].

Studies have shown that there is a significant interaction between diabetes and obesity on the risk for concentric cardiomyocyte hypertrophy [31,32,33]. The Strong Heart Study conducted on Native Americans revealed that higher LV mass and wall thickness were evident in cases with diabetes, explaining the association between diabetes and LVH. Elevated extracellular matrix (ECM) deposition leads to fibrotic changes in the heart, resulting in impaired LV relaxation and inefficient LV contraction [34,35].

## 5. Pathophysiological Anomalies Underlying DCM

### 5.1. Insulin Resistance and the State of Hyperglycemia

High levels of blood sugar or hyperglycemia are the primary cause of vascular complications in T2DM, which includes DCM [16]. DCM and heart failure are two distinct conditions that coexist with each other, and their respective causes and outcomes are dependent on each other, resulting in a bidirectional effect that explains the harmful changes that occur in the heart in DCM. In the case of type 2 diabetes, over time, there is a decrease in pancreatic β-cell mass, which leads to hyperglycemia, hyperlipidemia, a reduction in myocardial glucose transporter expression, and a decrease in glucose uptake. These metabolic disturbances act as the central catalyst for inducing maladaptive responses in DCM [4,36].

The heart can use two metabolic energy sources, glucose and FFAs, which it switches between depending on the demand and status of the physiological system. CD36, also known as scavenger receptor B2, allows fatty acids to enter cardiomyocytes, while GLUT-4 allows glucose to enter. In the fasting state, the heart uses FFAs as the metabolic substrate for energy production, whereas in the postprandial state, it uses glucose as the major substrate [4,37].

Insulin resistance is often associated with elevated fatty acid uptake mediated by the membrane glycoprotein CD36. Additionally, AMP-activated protein kinase (AMPK) activation is diminished. Those changes contribute to reduced expression and translocation of GLUT-4. Oxidative stress, led by increased production of ROS, is usually associated with persistent hyperglycemia and hyperinsulinemia, which leads to the drifting of glucose metabolism toward the hexosamine biosynthetic pathway (HBP) and the polyol pathway. The onset of the HBP leads to the generation of the nucleotide sugar uridine diphosphate-N-acetylglucosamine (GlcNAc) that acts as a substrate for glycosylation. The activated HBP and the polyol pathway are responsible for augmented biosynthesis of AGEs [38,39].

The accumulation of AGEs leads to interactions with RAGE, remodeling of the extracellular matrix with collagen elastin cross-linkage, activation of the NF-κB signaling pathway, production of TGF-β and proinflammatory cytokines, enhanced cardiac oxidative stress by ROS production, reduction in Ca^2+^ reuptake into the sarcoplasmic reticulum, myosin heavy chain (MHC) shift from α-MHC to β-MHC, and upregulation of atrial natriuretic peptide and BNP. These changes result in myocyte enlargement and myocardial fibrosis [39,40]. The overproduction of free radicals in mitochondria leads to reduced myocardial contractility and subsequent myocardial fibrosis. The oxidative stress also leads to apoptosis of cardiomyocytes and damage to cellular DNA. This oxidative stress-induced DNA damage activates several downstream cascade mechanisms that are involved in DNA repair and programmed cell death through increased levels of poly (ADP-ribose) polymerase enzymes (PARP). This shifts the pathway of glucose metabolism from its usual glycolytic pathway to an alternative pathway resulting in the generation of different mediators that cause diabetes-induced cellular damage [40].

The imbalance of tissue inhibitors of metalloproteinases (TIMPs) and matrix metalloproteinases (MMPs) is thought to be associated with inflammatory and fibrotic issues during hyperglycemia and insulin resistance. Conversely, the development of early glycosylation products can be the reason for the disparity of MMP/TIMP balance. TIMPs, in association with the MMPs, play a pivotal role in maintaining the stability and function of the ECM. Fibrotic changes are a very frequent manifestation of disruption in this balance. TIMP3 has a high affinity for the proteoglycans in the ECM and a broad range of substrates, including all MMPs [41]. Reduction in TIMP3 levels is observed in various cardiovascular diseases. Additionally, studies have indicated improvement in the disease condition has been achieved by TIMP3 replenishment, suggesting a putative role of TIMP3 in cardiovascular diseases. Stohr et al. investigated the effects of TIMP3 on cardiac energy homeostasis during increased metabolic stress conditions in TIMP3 knockout C57BL/6 mice [42]. Accumulation of neutral lipids was seen in the hearts of TIMP3 knockout mice along with elevated markers of oxidative stress. Apelin was identified to modulate metabolic defects, indicating that TIMP3 regulates lipid metabolism as well as oxidative stress response via apelin [42]. In summary, the role of insulin resistance in the pathophysiology of DCM is depicted in Figure 3.

### 5.2. Altered Insulin Signaling Cascades

A key pathophysiological defect associated with DCM is the abnormality in insulin metabolic signaling mechanisms in the heart induced by hyperglycemic states. In insulin-resistant states there exists an imbalance between the metabolic and growth effects of insulin signaling following which the MAPK pathway of insulin signal transduction predominates. This imbalance thereby modulates increased phosphorylation of serine of IRS-1, resulting in impaired PI3K signaling and Akt stimulation. The mTOR-S6 kinase 1 pathway forms to be an initiating pathway associated with impaired signaling cascades and is highly regulated by over-nutrition (nutrient sensor) and causing increased S6 kinase 1 phosphorylation, subsequently promoting insulin resistance in the heart, liver, adipose tissues, and skeletal muscle. This impairs PI3K involvement and PKB (Akt) stimulation, which, in turn, causes a reduction in glucose uptake. In addition, an altered insulin metabolic signaling state causes inhibition of insulin-stimulated coronary endothelial NOS activity and NO generation in the heart, which also increases and elevates intracellular Ca^2+^ levels in cardiomyocytes through the cGMP/PKG signaling pathway. These recurrent abnormalities progress to cause diastolic dysfunction and cardiac stiffness [22,23].

### 5.3. Hyperinsulinemia

Cardiomyocyte hypertrophy is augmented by the condition of hyperinsulinemia, which is modulated at the transcriptional level in DM. Multiple transcriptional factors are activated through several epigenetic and genetic alterations that regulate extracellular and cellular protein expression resulting from hyperinsulinemia. Such transcription factors stimulate the deposition of ECM proteins and the development of cardiomyocyte hypertrophy, which further results in myocardial fibrosis in DM [22,43].

### 5.4. Altered Metabolic Cascades: Lipotoxicity and Glucotoxicity

As β-cell mass progressively decreases in hyperglycemia, there is a deterioration in the function of β-cells that are chronically exposed to hyperglycemia, leading to a detrimental effect of glucotoxicity. This results in protein glycation reactions induced by hyperglycemia and glucotoxicity, which lead to increased levels of AGEs. These AGEs contribute to changes in the mechanical features of the ECM, such as increased connective tissue remodeling and cross-linking of collagens and laminins, leading to ventricular remodeling changes that play an integral role in the development of myocardial fibrosis in DCM. The associated activation of AGE/RAGE signaling also contributes to some maladaptive expression of genes that are augmented through MAPK and JAK pathways, thereby elevating levels of matrix proteins in cardiac and vascular tissues [23,44,45].

Another major consequence of insulin resistance is a rise in levels of FFAs, which leads to hyperlipidemia and lipotoxicity (lipid accumulation). Decreased glucose oxidation leads to augmented FFA oxidation, which contributes to aberrant myocardial morphology. In insulin-resistant states, increased FFA oxidation causes a rise in levels of ROS and promotes myocardial mitochondria dysfunction, which then accentuates mitochondrial uncoupling and elevated oxygen consumption, leading to decreased myocardial efficiency and relative cardiac ischemia. High levels of FFA also activate multiple genes transcription through PPAR-α myocyte expression and increase mitochondrial FFA transport and oxidation [46,47].

Emerging evidence reveals a correlation between high levels of triglycerides (TG) and myocardial damage. Hypertriglyceridemia hinders myocyte metabolism and contractility and triggers peripheral insulin resistance. It promotes cardiomyocyte apoptosis via increased ROS production and endoplasmic stress and contributes to diabetic myocardium remodeling [48,49,50]. By disrupting insulin metabolic signaling, certain lipid metabolites contribute to the exacerbation of DCM. The accumulation of diacylglycerols in the plasma membrane can activate protein kinase C (PKC), resulting in insulin resistance and a reduction in NO generation. Ceramides can directly activate PKC, inhibiting GLUT4 translocation and glucose uptake [23].

The epicardial adipose tissue (EAT) exerts mechanical stress on the myocardium and is thought to exacerbate atherosclerosis, cardiac remodeling, and heart failure by the release of inflammatory mediators and lipid metabolites. High epicardial fat-bearing T2DM patients are especially vulnerable to the negative effects of this adipose burden. This fat deposit is thought to be an important connecting link between diabetes, obesity, and cardiovascular ailments. EAT-derived proinflammatory adipocytokines including TNF-α, IL-1, and IL-6 are thought to exacerbate the inflammatory responses in the myocardium [51]. Emerging evidence revealed that EAT has lower GLUT4 expression than subcutaneous adipose tissue, which may signify worsening cardiac insulin resistance. Abnormal GLUT4 of EAT leads to increased cardiac insulin resistance. Glucotoxicity along with mitochondrial dysfunction, inflammatory factor release, and oxidative stress aggravate cardiac dysfunction [51,52].

### 5.5. Altered Neurohumoral Activation: Cardiac Autonomic Neuropathy in DCM

An established relationship exists between the progression and development of DCM and nervous system activation states with an early-stage characteristic of reduced parasympathetic activity and a relatively imbalanced. Moreover, a higher sympathetic activity has been demonstrated as the connecting link between autonomic dysfunction and abnormal cardiac functioning in the diabetic population [53]. Cardiac autonomic neuropathy (CAN) is a concomitant and chronic complication arising due to persistent hyperglycemia, and its associated abnormalities generally include altered heart rate, heart rhythm, vascular hemodynamic abnormalities, and anomalous cardiac structure and function. Stimulation in sympathetic nervous system activity causes activation and a rise in adrenergic β-1 receptor signaling and expression cascades, which, in turn, promote cardiomyocyte hypertrophy, interstitial fibrosis, the impaired contractile function of the myocardium, and increased cardiomyocyte apoptosis. During the occurrence of heart failure, there are instances of reduced parasympathetic pathway activation accompanied by alteration of muscarinic receptor composition and density, and decreased acetylcholinesterase activity. The severity of CAN is correlated with diastolic dysfunction in diabetic conditions exhibiting increased peripheral resistance and reduced vascular elasticity in association with stress [3,54].

### 5.6. Altered RAAS: Neurohormonal Abnormalities

In diabetes-induced cardiac dysfunction, a significant role of RAAS exists (Figure 4). Hyperglycemia and insulin resistance cause increased activation of RAAS and overproduction of angiotensin II (AGT-II), thereby exhibiting different effects on cardiomyocytes such as increased vascular resistance and arterial pressure. The proliferation of cardiac fibroblast and cardiomyocyte hypertrophy is induced due to a direct effect of AGT-II on cell signaling, and the levels of intracellular AGT-II are higher in myocardial cells of diabetic populations. Activation of systemic RAAS is responsible for impaired insulin metabolic signaling. This effect is orchestrated by the increased mineralocorticoids, which, in turn, induce insulin resistance in the myocardium by activating of mTOR/S6 kinase 1 signaling pathway. It also induces alteration in the signaling cascades due to enhanced AGT-II type 1 receptor and mineralocorticoid receptor in the myocardium, which causes increased adaptive immune response processes and triggers leucocyte adhesion and macrophage infiltration. These collective abnormalities lead to the induction of cardiac fibrosis through growth and profibrotic signaling pathways, cardiac remodeling, and diastolic dysfunction [23,55].

### 5.7. Endothelial Abnormalities

Vasodilators such as nitric oxide (NO), prostacyclin, bradykinin, and endothelium-derived relaxation factor (EDRF) are major mediators, which regulate cardiovascular homeostasis. Glucotoxicity resulting from insulin resistance potentiates the state of dysfunctional coronary endothelial cells and alters the natural physiological properties of the endothelium, thereby causing increased capillary permeability, leukocyte adhesion, and reduced fibrinolysis. Simultaneously, the expression of the proinflammatory cytokines, tumor necrosis factor-alpha (TNF-α) is enhanced, which, in turn, upregulates expression of vascular and intercellular cell adhesion molecules and promotes adherence of monocytes. The rise in the expression of TNF-α reduces endothelial NOS expression and interferes with NO production [56,57].

In DCM, initial stages of insulin resistance, and under hyperglycemic conditions, there is an impeded vasodilation induced by NO, whereas EDRF-mediated vasodilation is preserved or enhanced to regulate normal vascular tone. However, in the later stages of DCM, microvascular dysfunction assumes prominence due to impaired NO and EDRF-mediated vasodilation [58,59].

### 5.8. Mitochondrial Maladaptive Role in DCM

The myocardial mitochondria critically orchestrate critical functions such as bioenergetics, cytoplasmic calcium levels, apoptosis and autophagy, and countering oxidative stress. The pathogenesis of several cardiac events involves cardiac mitochondrial dysfunction, and diabetic myocardial defects are associated with severe alteration in lipid metabolism and insulin resistance. In a diabetic heart, there is augmented β-oxidation of fatty acids, which, in turn, induces accumulation of toxic lipid metabolites leading to cardiac lipotoxicity, fibrosis, mitochondrial damage, diastolic dysfunction, and heart failure [60,61,62].

The mitochondrial uncoupling mechanism, which involves increased oxygen consumption without an accompanying rise in ATP production, thereby decreasing cardiac efficiency, is an additional defect following insulin resistance [63,64]. Moreover, hyperglycemia and its associated instances of myocardial fibrosis cause an increased formation of ROS and free radicals, thereby augmenting the levels of oxidative stress, abnormal gene expression, and altered signal transduction process, which further lead to increased cascades of apoptosis and necrosis. In diabetes, the upregulation of RAAS enhances oxidative damage, thereby activating cardiac cell death through programmed cell death mechanisms [38,65].

### 5.9. Dysfunctional Immune Responses

Activation of proinflammatory cells such as macrophages and T lymphocytes is manifested in DCM. In the states of central adiposity and insulin resistance, elevated FFAs contribute to inflammatory conditions within the visceral adipose tissue, thus promoting subsequent systemic and cardiovascular inflammatory cytokine expression of TNF-α, IL-6, and monocyte chemotactic protein 1. These inflammatory mediators promote cardiac oxidative stress and coronary artery dysfunction, culminating in cardiac remodeling and fibrosis in DCM. Furthermore, the state of hyperglycemia and rise in levels of FFAs lead to NF-κB-induced inflammatory cytokine production. Additionally, RAGE signaling increases the expression of proinflammatory cytokines, causing subsequent cardiac fibrosis [66,67]. This leads to suppression of tumorigenicity 2 (ST2) expression, which is a member of the IL-1 receptor family [68]. It is present in two isoforms, the ST2 ligand (ST2L) and the soluble form of ST2 (sST2). ST2L acts as a receptor for IL-33, while sST2 is a soluble receptor that circulates in the bloodstream. It has been revealed that the interaction between IL-33 and ST2L improves myocardial function and protects hearts by lowering cardiomyocyte hypertrophy, myocardial fibrosis, and apoptosis [69]. In contrast, sST2 competes with ST2L by avid binding to IL-33 resulting in the suppression of the IL-33/ST2L system. As a consequence, the cardioprotective effects of ST2L are compromised. Elevated levels of sST2 in the plasma of T2DM patients indicate the interplay between insulin resistance and cardiotoxicity that is mediated by sST2 [70].

### 5.10. Role of ERS, Impaired Calcium Handling, and Cardiomyocyte Mass Depletion in DCM

Insulin resistance-induced severe ERS, impaired calcium signaling, and accretion of misfolded proteins play a pivotal role in the progression of DCM [71]. Increased misfolding of proteins and induced ERS ultimately leads to increased cell apoptosis and autophagy through a Ca^2+^-dependent pathway, which involves inositol-requiring kinase-1 (IRE1) and protein kinase R-like endoplasmic reticulum kinase (PERK) pathways [72,73].

Additionally, a mechanistic benchmark in the progression of DCM lies in accordance with impaired calcium signaling along with increased levels of oxidative stress and lipotoxicity. Normal cytosolic calcium levels regulate several physiological cascade mechanisms such as cellular metabolism, muscle contraction, and cell signaling. The excitation–contraction coupling of cardiomyocytes involves calcium as the essential ion. In this condition, calcium enters cardiomyocytes cytoplasm through voltage-sensitive L-type Ca^2+^ channels, after which it attains depolarization of the sarcoplasmic reticulum and triggers the release of cardiac Ca^2+^ through the activation of ryanodine receptors (RyR). This released Ca^2+^ then binds troponin C and induces contraction. In contrast, during the cardiac relaxation process, Ca^2+^ reverts back into the sarcoplasmic reticulum through activation of the sarcoplasmic reticulum Ca^2+^ pump and Na^+^/Ca^2+^ exchanger [74,75] (Figure 5).

However, in DCM, this Ca^2+^ reuptake mechanism is impaired, which results in the prolongation of action potential and persistence in the depolarized state, causing significant associated diastolic dysfunction due to decreased cardiac Ca^2+^ efflux, elevated intracellular resting Ca^2+^, prolongation of intracellular Ca^2+^ decay, and slowed Ca^2+^ transients. The collective interactions of ER stress and impaired Ca^2+^ handling thereby promote apoptosis in cardiomyocytes along with fibrotic changes associated with increased deposition of collagen. Increased mitochondrial permeability to sarcoplasmic Ca^2+^ overload also contributes to apoptosis augmenting the transition to both diastolic and systolic cardiac dysfunction in DCM [71]. The overall pathophysiological anomalies underlying DCM are depicted in Figure 6.

## 6. Existing Therapeutic Approaches and Their Limitations in the Management of DCM

Due to its complex and multifactorial-dependent progression, the therapeutic management of DCM has remained challenging. Hyperglycemia is the primary predisposing factor responsible for the development and progression of DCM [22,43]. The stages of DCM are associated with severe alterations in the myocardium, which provides a potential approach to the diagnosis and maintenance of DCM through the detection of LV alterations and related changes in diabetic populations. However, it involves a string of subclinical characterizations and is present without any significantly overt symptoms of the disease during the early stages. Thus, a better understanding of the pathological pathways of DCM in diabetic patients is needed [8,22,23].

Lifestyle modifications, antidiabetic medications, and management of co-occurring phenomena such as hypertension, ischemic heart disease, dyslipidemia, and other cardiac complications are being implemented to minimize events. The wide array of factors involved in the progression of DCM indicates that various treatment regimens might be effective in preventing and managing the progression of DCM over time [23,43]. The existing conventional therapies for diabetes and hypertension, which involve proper glycemic control, RAAS inhibition, lipid-lowering agents, calcium channel blockers, maintenance of prediabetic conditions, and lifestyle modifications, have been implicated in the management and delaying of DCM. They regulate blood glucose levels and insulin secretion by antidiabetic agents, control multifactorial risks in cardiovascular abnormalities, and prevent heart failure over time [43,76].

Metformin improves glucose utilization, reducing instances of cardiac remodeling due to decreased hypertrophy and fibrosis. Sulfonylureas increase cardiac function by decreasing myocardial fibrosis and oxidative stress. PPARα agonists such as thiazolidinediones (TZDs) increase glucose utilization, lower lipid levels, decrease fibrosis/steatosis, and contribute to better cardiac functioning. Statins reduce the chances of myocarditis due to reduced inflammation, oxidative stress, and subsequent fibrosis [23,77,78].

However, the existing therapies have significant and long-term limitations in the treatment of DCM. Metformin therapy is associated with reduced absorption of folate and vitamin B12, leading to elevated plasma homocysteine levels, which significantly increase the risk of thrombotic events by interfering with normal platelet, clotting factor, and endothelial function. Sulfonylureas also pose a significant risk of cardiovascular events. The use of PPARα agonists correlates with a risk factor in heart failure due to its persistent dependence on plasma lipid reduction and recovery of dysfunctional cardiac activities. The adverse impact of insulin production, insulin sensitivity, and pronounced plasma lipid reduction due to statins produces an additional direction in the advent of heart failure and several other cardiovascular events. TZDs are involved in the elevation of low-density lipoprotein (LDL) cholesterol levels, which significantly correlates with the association of PPAR-β overexpression with higher chances of myocardial lipogenesis in the heart. This class of drugs is also associated with a decrease in both atrial natriuretic peptide (ANP) and brain natriuretic peptide (BNP) levels, thereby resulting in increased volume status. It causes an increase in vascular permeability when associated with increased VEGF expression, predisposed to develop edema [79,80,81,82,83,84,85].

Insulin therapy appears to be a vital component in the management of DCM due to its ability to completely reverse underlying molecular mechanisms of oxidative stress involved in DCM. However, this reversal theory is feasible only if structural damage induced by hyperglycemia is not severe and irreversible [86]. The negative metabolic effects of β-blockers, such as diminished glycemic control and increased insulin resistance, have led to reluctant use of them in the management of DCM [87].

In addition to the serious implications of these conventional therapies, there lies a significant chain of adverse reactions under their applications, thereby causing gastrointestinal disturbances, nausea, hepatotoxicity, diarrhea, abdominal pain, lactic acidosis hypoglycemia, and even weight gain chances. The mentioned theories and situations demand some more potential targets be worked on, as in case of DCM, which may further reduce the chances and instances of associated complications in diabetic patients.

## 7. Emerging Pharmacotherapies

### 7.1. Sodium-Glucose Cotransporter 2 (SGLT2) Inhibitors

The introduction of novel anti-hyperglycemic agents has provided more potential therapeutic targets and treatment regimens. One such target is the inhibition of SGLT2, which reduces renal tubular glucose reabsorption through a novel mechanism. This action results in a reduction in blood glucose levels without stimulating insulin release further from the β-cells. Consequently, it provides a favorable condition for regulating normal blood pressure and body weight maintenance [78]. SGLT2 transporters are expressed in the proximal convoluted tubule of the nephrons and have a significant contribution to tubular glucose reabsorption. However, in T2DM, the threshold for renal glucose reabsorption becomes dysregulated, leading to overexpression of SGLT2 and worsening hyperglycemia due to maladaptive responses. Inhibiting SGLT2 produces significant cardio-renal benefits, reducing the chances of cardiovascular complications and heart failure. The USFDA has approved canagliflozin, dapagliflozin, and empagliflozin as SGLT2 inhibitors for the treatment of DCM. Additionally, the FDA has approved four combination drugs, namely, canagliflozin/metformin (Invokamet^®^), dapagliflozin/metformin (Xigduo XR^®^), empagliflozin/metformin (Synjardy^®^), and empagliflozin/linagliptin (Glyxambi^®^). By inhibiting tubular glucose reabsorption, these agents increase urinary glucose excretion, leading to lower blood glucose levels. The correlation between SGLT2 inhibition and cardio-protective mechanisms emphasizes its ability to improve myocardial metabolism and substrate utilization, thereby reducing preload and myocardial stress through the limitation of cardiac fibrosis and necrosis. Moreover, it increases natriuresis by augmenting the downregulation of sodium-hydrogen exchanger 3 (NHE3) activity in the proximal tubule, thus restoring normal sodium homeostasis. These effects converge to improve ventricular loading conditions, thereby reducing blood pressure levels and improving vascular functions [41,87,88,89,90,91,92].

In instances of prevailing heart failure, myocardial NHE3 is upregulated. SGLT2 inhibition decreases its activity, promoting the restoration of mitochondrial Ca2+ handling in cardiomyocytes. SGLT2 inhibitors also improve the state of aortic stiffness and promote endothelial function by inducing vasodilation through activation of voltage-gated potassium channels and protein kinase G. Dapagliflozin has antifibrotic effects by suppressing collagen formation through the activation of M2 macrophages and inhibition of myofibroblast differentiation. Furthermore, SGLT2 inhibitors decrease adipokine and leptin levels, restore the balance between several other anti-inflammatory adipokines, and tend to reduce cardiac inflammation and fibrosis. Thus, the strategy of SGLT2 inhibition offers value as a promising therapeutic target in the management of DCM [83,84,85,86,87,88,89,90,91,92,93,94,95,96].

Two significant clinical studies, DAPA-HF (assessing dapagliflozin) and EMPEROR-Reduced (assessing empagliflozin), have evaluated the beneficial aspects of empagliflozin and dapagliflozin. In two separate phase 3, placebo-controlled trials, 10 mg once-daily doses each of empagliflozin and dapagliflozin were administered in patients with heart failure and an ejection fraction of 40% or less. The primary outcome of both trials exhibited the benefits offered by SGLT2 inhibition in reducing the combined risk of heart failure in patients. The potential of SGLT2 inhibition was indicated by a relative reduction of 26% in the combined risk of cardiovascular death or first hospitalization for heart failure and by a 25% decrease in recurrent hospitalizations for heart failure [97,98,99,100].

Patients with T2DM are at a higher risk of cognitive impairment and memory loss. Thus, hypoglycemic agents, which may ameliorate cognitive deficits, can be of additional benefit to diabetic patients [101]. Emerging evidence suggests that SGLT2 inhibitors are associated with improved brain function and retard the progress of memory deficits [102].

### 7.2. Glucagon-Like Peptide-1 (GLP-1) Mimetics

GLP-1 is the peptide hormone secreted from the entero-endocrine cells of the intestine and is secreted in response to the consumption of food. It is an incretin hormone, which aids in glycemic control by enhancing insulin release and increasing glucose sensitivity, which further causes a sustained HbA1c reduction and reduced glucagon expressions mediated by glucose-dependent pathways. This series of effects further augment the promotion of β-cell health, decreasing its chances of being destroyed in the further course of time. GLP-1 receptors are G-protein coupled receptors wherein adenylyl cyclase-mediated conversion of ATP into cAMP causes increased insulin secretion. GLP-1 receptor activation initiates a series of downstream cascades, which lead to gastric emptying retardation, appetite restraint, reduced levels of plasma lipoprotein, and decreased the level of blood pressure [102]. During the diabetic condition, significantly reduced levels of GLP-1 lead to impaired insulin activity and progressive insulin resistance. Liraglutide and semaglutide are the long-acting GLP-1 analogs, which are immune to cleavage by dipeptidyl peptidase-4 (DPP-4), thereby producing extended in vivo action, resulting in beneficial effects in T2DM patients. Along with being beneficial in diabetes and obesity, liraglutide has been proven to reduce systemic inflammatory conditions as indicated by decreased levels of TNF-α, interleukins and CD163. GLP-1 agonists promote insulin release only in hyperglycemic conditions, thus lowering the risk of hypoglycemia and thereby reducing the chances of associated cardiovascular complications [103,104].

Exchange protein activated by cyclic-AMP (Epac 2) is a guanine nucleotide exchange factor involved in cAMP-mediated signal transduction. GLP-1 receptor agonism has demonstrated relative activation of Epac-2, which subsequently induces increased ANP secretion and troponin I phosphorylation, resulting in increased myocyte contractility. Infusion of exogenous GLP-1 or treatment with GLP-1 receptor agonists through modulation of GLP-1 signaling cascades has shown a reduction in infarct dimensions, improved cardiac function, and improvement of left ventricular systolic functions. Thus, the cardioprotective effects mediated by GLP-1 analogs via receptor mechanisms in the heart and the endothelium has designated GLP-1 as a potential line of treatment in DCM [22,41,104,105,106,107,108,109,110,111].

Two large phase 3 Cardiovascular Outcomes Trials (CVOTs), SUSTAIN 6 and PIONEER 6, investigated the beneficial effects of subcutaneous and oral semaglutide in patients with T2DM and high cardiovascular risk. The SUSTAIN 6 trial was designed to assess the efficiency of semaglutide in terms of cardiovascular safety in patients with T2DM. This randomized, double-blind, placebo-controlled, parallel-group trial was conducted on 3297 patients with T2DM. The patients who were on a standard regimen for glycemic control received semaglutide (0.5 mg or 1.0 mg) once weekly for 104 weeks. The primary outcome was occurrence of cardiovascular failure, nonfatal myocardial infarction (MI), or stroke, the occurrence of which was 6.6% in the semaglutide group and 8.9% in the placebo group. Approximately, 2.9% of the patients receiving semaglutide had nonfatal MI. Similarly, nonfatal stroke occurred in 1.6% of patients. Additionally, worsening of nephropathy was reduced in the semaglutide group [112].

The PIONEER 6 trial was a different randomized, double-blind, placebo-controlled trial in which a total of 3183 patients were randomly assigned to receive oral semaglutide (14 mg) or placebo. The objective of PIONEER 6 was similar to that of SUSTAIN 6, the key difference being that PIONEER 6 was purely event-driven and did not involve duration-driven effects like the SUSTAIN 6 trials. Semaglutide administration resulted in a 21% reduction in cardiovascular effects, which was driven by fewer cardiovascular deaths and nonfatal strokes [113]. Based upon the successful results achieved by these trails, the FDA has approved the use of semaglutide in T2DM patients with known cardiovascular diseases.

Diabetic nephropathy also constitutes a significant disease burden of T2DM disease progression. GLP-1 mimetics and SGLT2 inhibitors have been shown to have positive effects on kidney function in individuals with T2DM [114]. A randomized single-blind study compared the effects of empagliflozin and liraglutide and their combination arterial stiffness in 62 adult patients with T2DM. It was observed that systemic vascular resistance and lipoprotein(a) levels improved to a greater extent with liraglutide than empagliflozin, along with improvement in BMI, body and visceral fat, blood pressure, and HbA1c. However, no improvement in arterial stiffness indices was observed with empagliflozin or liraglutide or their combination in this study [115].

### 7.3. DPP-4 Inhibitors

The activity of GLP-1 analogs in the maintenance and regulation of insulinotropic effects has provided additional insights into the cardio-protective mechanisms, which can be essential in the management of DCM. However, the existence of the enzyme DPP-4 as a key regulator in the breakdown of incretins has produced a challenging situation. Thus, the levels of gut hormones become significantly reduced and further lead to complications in the hyperglycemic state. DPP-4 elicits exopeptidase activity against several peptide hormones and chemokines such as BNP, NPY and substance P, thus regulating several other vascular abnormalities, inflammation, and cell survival. The DPP-4 inhibitors such as sitagliptin and saxagliptin prolong the activity of GLP-1, gastroinhibitory peptides, and other incretins such as peptide tyrosine and oxyntomodulin have had a remarkable impact on the renal and cardiovascular system. Takahashi and colleagues demonstrated the cardioprotective effects of vildagliptin in a murine heart failure model by inducing a transverse aortic constriction in C57BL/6J mice, simulating pressure-overloaded cardiac hypertrophy and heart failure. Vildagliptin was found to reduce myocardial apoptosis and fibrosis along with improving cardiac function in mice with transverse aortic constriction. In addition to this, the expression of DPP-8 and DPP-9 in the sarcoplasm of cardiomyocytes produces a direct impact on the cellular homeostasis and energy metabolism via cytosolic calreticulin and adenylate kinase 2, thus affecting myocardial function. Therefore, its inhibition may also be explored clinically [116,117,118,119].

### 7.4. Neprilysin Pathway Inhibitors

Neprilysin has received significant attention during the last few years on account of its underlying role in heart failure. It belongs to the family of metallopeptidases (M13) and is a membrane-bound, zinc-dependent metallopeptidase. Very high neprilysin concentration is seen at the proximal tubules of the nephrons. It cleaves and catalyzes the degradation of several vasoactive peptides, natriuretic peptides, adrenomedullin, angiotensin I and II, substance P, bradykinin, etc. In diabetes, there exists a state of RAAS upregulation, and its modulation comprises a significantly central role in the progression of the complications associated with heart failure. The natriuretic peptide system produces a counter effect and regulates the detrimental effects caused by the upregulation of RAAS, modulating the autonomic nervous system and thereby rendering beneficial effects in heart failure prognosis. Activation of RAAS, vasopressin, and the sympathetic nervous system leads to increased ventricular preload, afterload, and elevated wall stress, thereby causing the production of pre-pro BNP, which is further cleaved to BNP and N-terminal pro-BNP (NT-pro-BNP), which is a physiologically inactive form. In contrast, atrial stretching ultimately leads to the production of pre-pro-atrial or A-type natriuretic peptide and further ANP, which has a similar biological characteristic to BNP in promoting natriuresis and vasodilation [120,121,122]. Neprilysin plays a key role in the degradation of these peptides and hence complicates the underlying mechanism in cardiovascular events. The rationale for implementing neprilysin inhibitor therapy is to elevate the levels of circulating endogenous peptides levels and hence achieve vasodilation as well as natriuresis by persisting longer retention time [123]. Racecadotril and ecadotril were the initial neprilysin inhibitors that were successful in attenuating natriuresis. However, studies have also claimed that neprilysin cleaves angiotensin II, which potentially counteracts the actions of many endogenous peptides. This situation demanded a solution that can implement predominant neprilysin inhibition [123]. The theory of dual neprilysin and ACE inhibition has been then implicated, which led to a dual blockade of RAAS as well as the natriuretic peptide system. LCZ696 is a novel dual-acting angiotensin receptor-neprilysin inhibitor (ARNI) being investigated for its potential in heart failure with reduced ejection fraction (HFrEF). It is a combination of the angiotensin II receptor blocker valsartan and the neprilysin inhibitor sacubitril. Sacubitril is further hydrolyzed to form a potent metabolite, LBQ657 [124,125,126].

Three clinical studies have supported the use of neprilysin inhibitors in the management of cardiac complications in T2DM patients and demonstrated beneficial metabolic effects with ARNI. The PARADIGM-HF trial (Prospective Comparison of ARNI with ACEI to Determine Impact on Global Mortality and Morbidity in Heart Failure) was a very significant study that involved analysis of patients with T2DM and heart failure. Greater reduction in HbA1c was demonstrated by long-term ARNI treatment. It also resulted in reducing the requirement of oral hypoglycemic or insulin therapy, compared with an ACE inhibitor alone [96,127,128]. In another recent clinical study, ARNI improved glycemic control and insulin sensitivity in individuals with T2DM and/or obesity [129]. After 8 weeks of treatment of obese hypertensive patients with ARNI, there was significant increase in the insulin sensitivity index. This was significantly higher in patients treated with ARNI as compared to amlodipine.

### 7.5. SERCA as a Promising Target Therapy

The membrane-bound active transporter SERCA (sarcoplasmic/endoplasmic reticulum Ca^2+^ ATPase) carries out the translocation of cytosolic Ca^2+^ into the lumen of sarcoplasmic reticulum. This process is hampered in the diabetic myocardium, which contributes to diastolic dysfunction and hence subsequent DCM. Additionally, dysfunctional myocyte calcium homeostasis induces cardiomyocyte death finally converging to DCM. Thus, improved activity of SERCA presents itself as a potential target to improve myocardial performance [130,131].

Exendin-4 is a 39 amino acid agonist of GLP-1 receptor that has been found to reduce apoptosis in neonatal rat ventricular cardiomyocytes subjected to high glucose present in the culture medium. This antiapoptotic activity is blocked by the GLP-1 receptor antagonists. In an interesting observation, it has been reported that exendin-4 protects cardiomyocytes from H_2_O_2_-induced cell death, which was attributed to reduced ER stress. This in turn was associated with reduced expression of glucose-regulated rotein-78 (GRP78). GRP (78 kDa protein), an ER-resident protein chaperone, acts as a master regulator of many unfolded protein response signaling branches. The fact that exendin-4 treatment selectively protected cardiomyocytes from thapsigargin (a noncompetitive SERCA inhibitor)-induced death suggests that exendin-4 attenuates thapsigargin-mediated inhibition of the SERCA2a. Along similar lines, high glucose levels act to attenuate SERCA2a function, but exendin-4 treatment prevented this reduction. These findings suggest that GLP-1 receptor activation could attenuate high glucose-induced cardiomyocyte apoptosis by mechanisms involving reduced ER stress and by improved SERCA2a activity, which further boosts the use of GLP-1 receptor-based therapies as treatments for DCM [43,132,133].

A preclinical study investigating the role of erythropoietin (EPO) has been found to reduce cardiac dysfunction by suppressing ER stress in diabetic rats. The animals in the EPO-treated control and diabetic groups were administered recombinant human EPO (1000 U/kg body weight) once per week for 12 weeks, and the expression of GRP78 and SERCA2a was assessed. Treatment with EPO attenuated the increase in GRP78 and increased SERCA2a expression in diabetic rats. Furthermore, EPO administration inhibited hyperglycemia-provoked cardiomyocyte apoptosis. This study revealed that EPO treatment could improve the parameters of cardiac function following high glucose-induced injury by suppressing ER stress and inducing SERCA2a expression, thus emphasizing the protective role increased activity of SERCA in management of DCM [134].

### 7.6. Application of Adrenomedullin (ADM) as a Potent Endogenous Vasodilator

ADM is a ubiquitous peptide primarily involved in vasodilatation of both arterial and venous blood vessels. It is also believed to preserve endothelial integrity. In addition, ADM inhibits the RAAS system. ADM signaling occurs through heterodimeric receptor complexes called “ADM receptors”, which are composed of a calcitonin receptor-like receptor (CRLR) and receptor activity-modifying proteins (RAMP2 or RAMP3). It results in the generation of cAMP and NO, causing vasodilation and possible inhibition of oxidative stress, hypoxic injury, and angiogenesis, leading to decreased LV end-diastolic pressure and increased cardiac output. A mature moiety of ADM contains an intramolecular disulfide bond and possesses some additional similarity to the calcitonin family such as CGRP, amylin, and calcitonin. It also promotes reduction in cardiac hypertrophy and fibrosis. ADM, when combined with human ANP, reduces oxidative stress, mean arterial pressure, and systemic and pulmonary vascular resistance without altering heart rate as given through studies that were performed on patients with acute decompensated and chronic heart failure. In addition to this, it also helps to inhibit proliferation and collagen production in cardiac fibroblasts, thus proving its characteristics to promote beneficial effects on the kidney [135,136].

### 7.7. Antioxidant Strategies in the Management of DCM

Elevated oxidative stress contributes to a great extent to the progression of pathological changes in DCM [7,20,137]. Several studies have highlighted that mitigation of oxidative stress could retard the progression of DCM. Some of the interesting studies that could help in curating therapeutic agents useful in the management of DCM are mentioned in Table 1.

### 7.8. Glucose-Dependent Insulinotropic Polypeptide (GIP) Agonists

GIP is responsible for amplification of insulin secretion after food intake along with GLP-1. Tirzepatide, an acylated peptide, is the first dual GIP-GLP-1 receptor coagonist approved for the treatment of T2DM. Five clinical trials in patients with T2DM have shown that tirzepatide reduces both HbA1c and body weight. An important point to highlight is that tirzepatide was found to be significantly more effective in reducing HbA1c and body weight than semaglutide. Tirzepatide was also found to improve insulin sensitivity and insulin secretory responses to a greater extent than semaglutide [159]. GIP-based therapeutics may be useful in DCM by ensuring glycemic control and restoring insulin responsiveness.

### 7.9. Imeglimin

Imeglimin is a novel oral agent approved for the treatment of T2DM [160]. It is a mechanism of action that is unique and involves dual effects of enhancing insulin action and reversing pancreatic β-cell dysfunction. Imeglimin augments glucose-stimulated insulin secretion and induces improvement of β-cell mass. Very interestingly, it has been found to reverse insulin resistance also by inhibiting hepatic glucose output and enhancing insulin signaling in the liver and skeletal muscle [161]. Imeglimin is believed to correct mitochondrial dysfunction resulting in reduced oxidative stress and prevention of mitochondrial permeability transition pore opening (implicated in preventing cell death); thus, it could serve as a potential therapeutic agent to treat DCM.

### 7.10. Miscellaneous Targets

Since the progression of DCM is multifactorial, modulation of several signaling pathways has shown amelioration of diabetes-associated pathological cardiac changes. These diverse targets involve metabolic pathways, zinc homeostasis, ubiquitin ligase, etc., and are summarized in Table 2.

## 8. MicroRNAs (miRs) as Potential Biomarkers in DCM

In recent times, miRs have emerged as promising biomarkers for the early detection of DCM, presenting a new avenue for the diagnosis and treatment of this condition. These tiny, noncoding RNA molecules play a critical role in regulating gene expression, and in diabetic individuals, their levels are altered in cardiomyocytes, leading to changes in the plasma levels of significant cardiac biomarkers [176]. Among different miRs, miR-21, miR-24, miR-30d, miR-34a, miR-141, miR-195, miR-142-3p, miR-199a-3p, miR-206, miR-208a, miR-216a, miR-221, miR-223, miR-301a, miR-320, miR-451, miR-483-3p, miR-499-3p, miR-700, and miR-705 remain upregulated in DCM patients. These could serve as potential biomarkers for diagnosis of DCM [177]. On the other hand, miR-1, miR-9, miR-20a, miR-23b, miR-29, miR-30c, miR-133a, miR-143, miR-144, miR-150, miR-181a, miR-220b, miR-373, miR-378, and miR-499 have been found to be downregulated in DCM [177]. In general, suppression of the upregulated miRs and activation of downregulated miRs in DCM milieu could serve as a therapeutic approach. Among aforementioned miRs, miR-223, miR-1, miR-499, and miR-133a have been shown to be involved in the post-translational modulation of several kinases in diabetic conditions [163,176,178]. In particular, miR-223 (remains upregulated in DCM) plays a crucial role in the pathogenesis of DCM. Recently, Xu and colleagues reported that inhibition of miR-223 improved the morphological anomalies and degree of fibrosis in myocardial tissues of DCM model rats [179]. The results demonstrated that inhibiting miR-223 suppressed the activation of NLRP3 inflammasome and improved myocardial fibrosis and apoptosis in the murine model of DCM. In contrast, upregulating miR-1, miR-499, and miR-133a, which remain downregulated in DCM, could be activated as a viable therapeutic approach to reduce DCM.

## 9. Conclusions and Future Directions

In this review, we have highlighted the unique nature of diabetic cardiomyopathy, which is characterized by a combination of subtle structural and functional changes that progress silently over time. This complication is associated with metabolic and molecular disturbances that ultimately lead to severe consequences such as early diastolic impairments, myocardial dysfunction, tissue fibrosis, and heart failure. The co-occurrence of DCM with chronic hyperglycemia, high blood pressure, dyslipidemia, and elevated oxidative stress further complicates the situation, making early detection crucial to minimize its severity.

While conventional therapies for diabetes, hypertension, and lipid-lowering agents have been used to restore normal cardiac functioning, the emergence of cellular and molecular perturbations in DCM has made the situation somewhat urgent. The limitations of existing mainstream therapies demand a more precise treatment regimen that can produce beneficial results while minimizing the underlying pathophysiology involved in DCM. To this end, modulation of several underlying pathways by various classes of compounds such as SGLT2 inhibitors, GLP-1 receptor analogs, α-glucosidase inhibitors, and antioxidants are being used to prevent or minimize the gradual progression of heart failure and ongoing treatment challenges. These potential treatment strategies provide a precise direction in the treatment of DCM by targeting specific pathways responsible for the onset of structural and functional alterations in the diabetic heart.

## Figures and Tables

**Figure 1 biomedicines-11-01126-f001:**
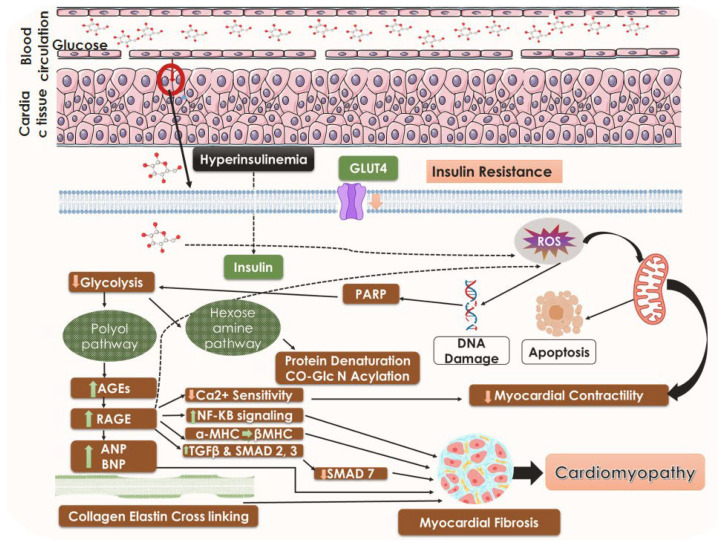
A general mechanistic understanding of DCM. Black arrows represent downstream events, black dotted arrows also represent downstream events however several steps are involved, the arrows pointing upward represent upregulation, and the arrows pointing downward represent suppression. GLUT 4, glucose transporter; AGEs, advanced glycation end products; RAGE, receptor for advanced glycation end products; ANP, atrial natriuretic peptide; BNP, brain natriuretic peptide; TGFβ, transforming growth factor-β; SMAD, suppressor of mothers against decapentaplegic; MHC, myosin heavy chain.

**Figure 2 biomedicines-11-01126-f002:**
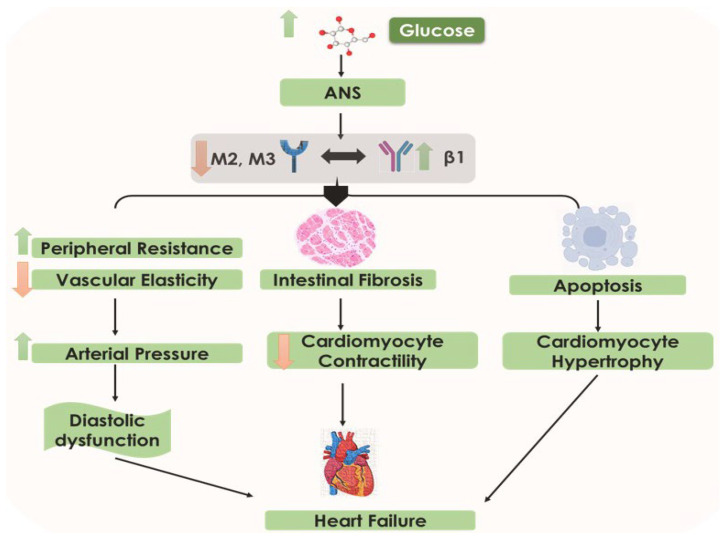
The sequence of events in diabetic heart failure. Increased glucose levels cause an increase in arterial pressure, decrease in cardiomyocyte contractility, cardiomyocyte hypertrophy, diastolic dysfunction, and heart failure. Arrows represent downstream events.

**Figure 3 biomedicines-11-01126-f003:**
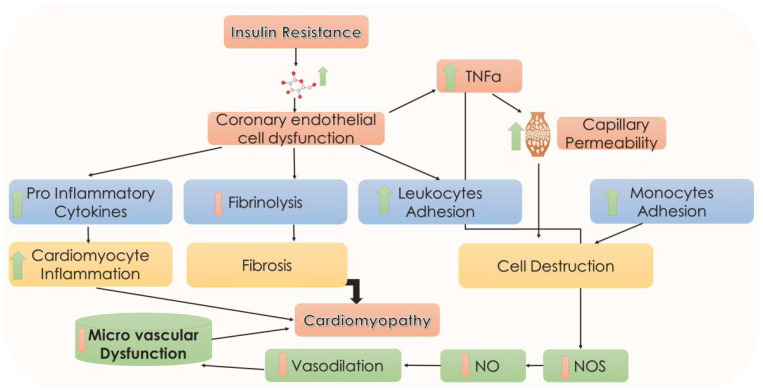
Insulin resistance and DCM. Coronary endothelial cell dysfunction may cause an increase in proinflammatory cytokines, TNFα, leukocytes, and monocyte adhesion and a decrease in fibrinolysis. All these sequence events may cause a decrease in microvascular dysfunction in the heart. Arrows represent downstream events. NO, nitric oxide; NOS, nitric oxide synthetase; TNFα, tumor necrosis factor-α.

**Figure 4 biomedicines-11-01126-f004:**
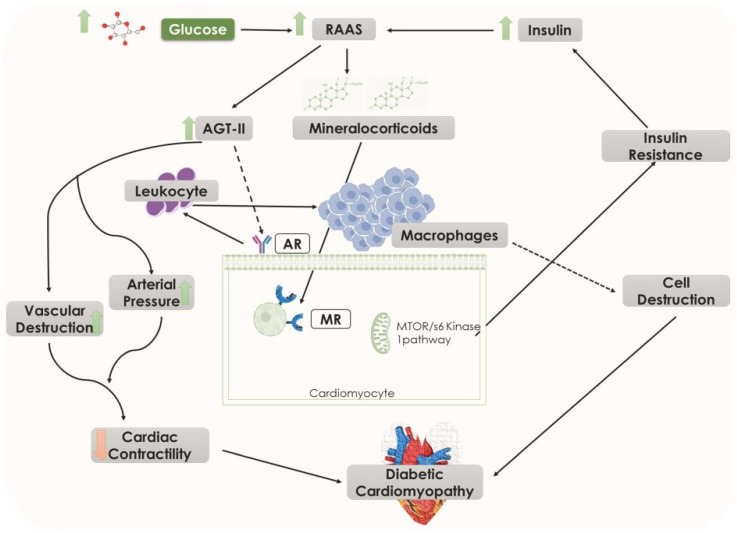
Illustrating the role of increased glucose levels in diabetic cardiomyopathy through elevated RAAS pathway. Black arrows represent downstream events and black dotted arrows also represent downstream events however several steps are involved. AGT-II, angiotensin II; AR, angiotensin receptor; MR, mineralocorticoid receptor; RAAS, renin–angiotensin–aldosterone system.

**Figure 5 biomedicines-11-01126-f005:**
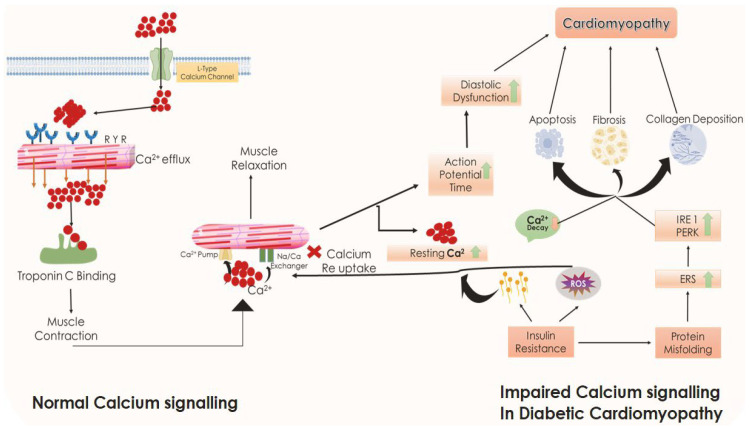
Comparison of normal calcium signaling with diabetic cardiomyopathy calcium signaling. Black arrows represent downstream events. ERS, endoplasmic reticulum stress; IRE1, inositol-requiring kinase-1; PERK, protein kinase R-like endoplasmic reticulum kinase; ROS, reactive oxygen species.

**Figure 6 biomedicines-11-01126-f006:**
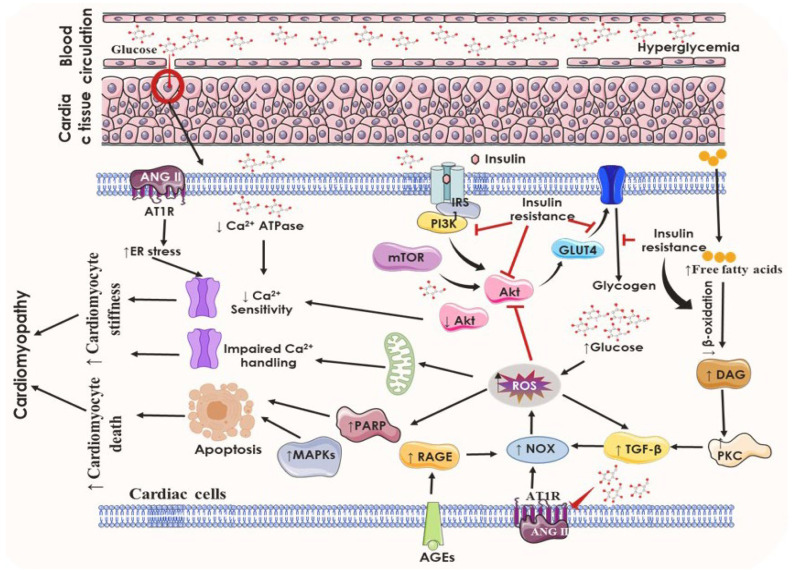
Overall pathophysiological anomalies underlying diabetic cardiomyopathy. Black arrows represent downstream events and red lines represent suppression. AGEs, advanced glycation end products; AGT II, angiotensin II; ATP, adenosine triphosphate; BNP, brain-derived natriuretic peptide; DAG, diacyl glycerol; ER stress, endoplasmic reticulum stress; GLUT, glucose transporters; IRS-1, insulin receptor substrate 1; MAPK, mitogen-activated protein kinases; mTOR, mammalian target of rapamycin; PARP, poly (ADP-ribose) polymerase enzymes; PI3K, phosphoinositide 3-kinases; PKC, protein kinase C; RAGEs, receptor for advanced glycation end products; ROS, reactive oxygen species; TGF, transforming growth factor.

**Table 1 biomedicines-11-01126-t001:** Antioxidant strategies in the management of DCM.

Sl. No.	Agents	Mechanisms	References
1	Coenzyme Q10	It elicits protection for cardiomyocyte function by augmenting antioxidant properties. It reduces cardiac inflammation, fibrosis, and hypertrophy induced by T1DM and T2DM.	[138,139]
2	Catalase	Upregulation of catalase causes improvement in the cardiac morphology, mitochondrial, and myofibrillar characters and cardiomyocyte contractility with a significant reduction in the levels of ROS. It has been found to ameliorate diabetes-induced autophagy by increasing NF-κB activity.	[61,140]
3	Thioredoxin	Thioredoxin 2 acts like a mitochondrial antioxidant that offers protection against oxidative stress. Overexpression of thioredoxin 2 has been reported to reduce high glucose-induced mitochondrial oxidative damage along with decreasing expression of ANP and BNP. Loss of thioredoxin 2 has been found to induce cardiomyocyte hypertrophy.	[141,142]
4	Edaravone	Edaravone inhibits fibrosis and cardiac apoptosis by activating Nrf2, NADP quinone oxidoreductase and heme oxygenase. Increased activity of sirtuin 1 and PGC-1α by edaravone has been reported. Additionally, it reduces apoptosis by increasing Bcl-2 expression and reducing Bax and caspase-3 expressions in cardiomyocytes.	[143,144]
5	Quercetin	Quercetin prevents cardiac remodeling by promoting Nrf-2 and inhibiting NF-κB signaling. It inhibits the RAAS pathway, decreases expression of TGF-β1, and subsequently reduces deposition of the ECM. Additionally, it modulates the sirtuin 3/PARP-1 pathway and inhibits cardiac hypertrophy.	[145,146]
6	Taxifolin	Taxifolin exerts an antifibrotic effect by inhibiting TGF-β/SMAD signaling. It has been found to improve diastolic dysfunction, ameliorate myocardium structure abnormality, and enhance endogenous antioxidant enzyme activities. It also reduces angiotensin II levels in the myocardium and inhibits NADPH oxidase activity. It has been found to prevent myocyte apoptosis by inhibiting caspase-3 and caspase-9 activation and restoring mitochondrial membrane potential.	[147,148]
7	Luteolin	Luteolin protects against DCM by inhibiting NF-κB and activating the Nrf2-mediated antioxidant responses. It inhibits TGF-β1, NOX4, and NOX2.	[149,150]
8	Kaempferol	Kaempferol inhibits NF-κB and endorses Nrf-2 activation. It prevents diabetes-induced cardiac fibrosis and apoptosis.	[151,152]
9	Szeto–Schiller peptide (SS31)	SS31 is a positively charged free radical scavenger that accumulates in the mitochondria and prevents diastolic dysfunction, myocardial fibrosis, and subsequent cardiac hypertrophy. SS31 has been reported to mitigate oxidative stress, autophagy, and ER stress.	[153,154]
10	Isorhamnetin	Isorhamnetin reduces cardiac fibrosis and hypertrophy. It improves insulin signaling and restores the arrangement of myofibrils. It upregulated Akt-2, microRNA (miR)-1, and miR-3163 expression in skeletal muscle and adipose tissue.	[155,156]
11	Sulphoraphane	Sulforaphane prevents DCM via the upregulation of Nrf2 and metallothionein. It also prevents ferroptosis and associated pathogenesis via AMPK-mediated NRF2 activation.	[157,158]

**Table 2 biomedicines-11-01126-t002:** Modulation of signaling pathways as promising targets.

Sl no.	Targets/Agents	Mechanisms	References
1	Cardiac PI3K (p110α) signaling pathway	Increased activation of the p110α pathway leads to improved diastolic dysfunction, cardiomyocyte hypertrophy, myocardial fibrosis, and programmed cell death in diabetic subjects, thus preserving ventricular function as well as augmenting cardiac structural remodeling. The beneficial effect of recombinant adeno-associated viral vectors carrying a constitutively active PI3K construct (rAAV6-caPI3K) in T2DM animals was studied. rAAV6-caPI3K gene-bearing animals showed a reduction in diabetes-induced cardiac remodeling by preventing cardiac fibrosis and cardiomyocyte hypertrophy. Additionally, LV reactive oxygen species and ER stress were reduced.	[22,162,163]
2	Long-chain 3-ketoacyl-CoA thiolase inhibitors	Trimetazidine (TMZ) is a competitive inhibitor of the long-chain 3-ketoacyl-CoA thiolase involved in the β-oxidation of fatty acids, which potentially improves myocardial metabolic or substrate utilization and reduces calcium overload, and ROS-induced cell injury. It reduces FFA utilization and enhances glucose oxidation along with decreasing insulin resistance. TMZ has been shown to be cardioprotective in DCM. TMZ treatment reciprocated LV dysfunction, cardiac hypertrophy, fibrosis, inflammation, and oxidative stress in the myocardium. Additionally, TMZ treatment inhibited diabetes-associated structural and functional alterations by inhibiting NADPH oxidase 2 and transient receptor potential channel 3. Furthermore, the administration of TMZ in an early stage of diabetes may inhibit the progression of DCM by inhibiting myocardial fibrosis and cardiomyocyte apoptosis and enhancing autophagy. TMZ has been found to reverse myocardial remodeling and reduce the deposition of collagen I and III content.	[164,165,166]
3	Metallothioneins (MTs)	MTs involved in the regulation of the intracellular zinc concentration have received significant attention due to the fact that supplementation with zinc has been found to be beneficial in the management of T2DM. MTs are considered a key regulator of zinc metabolism, and the redox process controlled by them causes the simultaneous release and regeneration of zinc-binding capacity. MTs have been found to be involved in the attenuation of oxidative stress by the scavenging of superoxide and hydroxyl radicals. MT prevents DCM and increases the expression of proteins associated with glucose metabolism. MT has been found to preserve Akt2 activity and cardiac function by inhibiting tribbles pseudokinase 3 (TRB3). Cardiac MT overexpression in Akt2 knockout mice was found to prevent pathological changes associated with DCM.	[167,168,169]
4	E3 ubiquitin ligase	The E3 ubiquitin ligases (E3s), the components of the ubiquitin-proteasome system, are believed to play a key role in the progression of DCM due to their involvement in cardiac hypertrophy, increased apoptosis, fibrosis, and altered insulin metabolism. In animal models of T2DM, it has been seen that there is an increase in the expression of E3s in the cardiac tissue, leading to proteasomal degradation of the insulin receptor and insulin receptor substrates, converging to a state of insulin resistance.	[170,171]
5	A novel curcumin analog, C66, in the management of DCM	C66[(2E,6E)-2,6-bis(2-(trifluoromethyl)benzylidene)cyclohexanone], a curcumin analogue, could be beneficial in the management of DCM. It has been reported to reduce hypertriglyceridemia in diabetic animals, along with reducing plasma and cardiac triglyceride levels. Additionally, it also inhibits Jun NH_2_-terminal kinase (JNK) and NF-κB activation in the heart.	[172,173]
6	Neuregulins (NRG)	NRG-1 is involved in cardiac damage adaptability, as well as maintenance of the shape of cardiomyocytes to limit apoptosis and increase cardiomyocyte proliferation. NRG-1 promotes mitochondrial homeostasis and stability and is considered to be an agent with the potential to ameliorate heart failure and other metabolic dysregulation and inflammation-related diseases such as obesity and T2DM.	[96,174,175]

## Data Availability

Not applicable.

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
