# Peer review of "Exploring the Complex Relationship between Diabetes and Cardiovascular Complications: Understanding Diabetic Cardiomyopathy and Promising Therapies"

_biomedicines, 2023, doi:10.3390/biomedicines11041126_

Round 1

Reviewer 1 Report

Very interesting review . Authors provide a wide scientific scenario on relathionship between diabetes and diabetic cardiomyopathy. Actually, heart failure, and in partuicular disatolic heart failure, contrary to normally perceived, is the most common cardiac complication ralated to diabetes. 

I have only some concerns.

1) In figures legend please specificy the meaning of each used acronym

2) The function of TIMP3 as a dual regulator of extracellular matrix remodelling and tissue inflammation, fibrosis/repair suggests a potential involvement of TIMP3 in cardiac tissue damage. I suggest to include this issue in the review using the following important paper which deserves to be included in reference list: PMID: 26500845

3) Finally, please also include the role of soluble ST2 in this review. It is a member of IL1 receptor-like family, it is secreted in response to myocardial strain and to IL1 stimulation. However, it is also linked to glucose abnormalities. Please discuss this issue, use this quality paper (PMID: 30259114) and include it among references.

Author Response

Comment 1: In figures legend please specify the meaning of each used acronym

Response: The suggested change has been implemented as per the recommendation.

Comment 2: The function of TIMP3 as a dual regulator of extracellular matrix remodelling and tissue inflammation, fibrosis/repair suggests a potential involvement of TIMP3 in cardiac tissue damage. I suggest to include this issue in the review using the following important paper which deserves to be included in reference list: PMID: 26500845

Response: Authors are thankful for this important comment. The manuscript has been modified and the role of TIMP3 has been critically along with the suggested reference.

Comment 3: Finally, please also include the role of soluble ST2 in this review. It is a member of IL1 receptor-like family, it is secreted in response to myocardial strain and to IL1 stimulation. However, it is also linked to glucose abnormalities. Please discuss this issue, use this quality paper (PMID: 30259114) and include it among references.

Response: Authors have included the potential roles of soluble ST2
(sST2) in this manuscript as per recommendation following the literature of the suggested manuscript and the article has been cited.

Reviewer 2 Report

It is of great interest to the reader because it provides a thorough literature review, discussion of the development mechanism and therapeutical drug of diabetic cardiomyopathy. However, I would like to make a few suggestions to further improve this review.

2.Epidemiology

Additional epidemiological information on heart failure in patients with type 2 diabetes would be desirable. Is the prevalence of heart failure in patients with type 2 diabetes decreasing with improved treatment? Or is it still as high as it used to be?

3.Progression of DCM

Recently, GIP formulations have become available as a treatment option for patients with type 2 diabetes; what is the reported association with GIP?

4. Cardiac structural and functional anomalies in DCM and  5. Pathophysiological anomalies underlying DCM

・Epicardial fat has recently received attention as a factor affecting diabetic cardiomyopathy, and its involvement should be discussed. How does reducing epicardial fat through drugs and lifestyle interventions affect DCM?

・Please mention GIP drugs as therapeutic agents for DCM.

・Please mention imeglimin as therapeutic agent for DCM.

Author Response

Comment 1: Epidemiology; Additional epidemiological information on heart failure in patients with type 2 diabetes would be desirable. Is the prevalence of heart failure in patients with type 2 diabetes decreasing with improved treatment? Or is it still as high as it used to be?

Response: The epidemiology section has been modified considering the suggestion.

Comment 2: Progression of DCM; Recently, GIP formulations have become available as a treatment option for patients with type 2 diabetes; what is the reported association with GIP?

Response: The section has been modified considering the valuable suggestion. The authors have included the possibilities of GIP targeting (GIP agonists) in DCM treatment in this manuscript.

Comment 3: Cardiac structural and functional anomalies in DCM and Pathophysiological anomalies underlying DCM. Epicardial fat has recently received attention as a factor affecting diabetic cardiomyopathy, and its involvement should be discussed. How does reducing epicardial fat through drugs and lifestyle interventions affect DCM? Please mention GIP drugs as therapeutic agents for DCM. Please mention imeglimin as therapeutic agent for DCM.

Response: Authors are thankful for these critical comments. The manuscript has been modified considering this recommendation and the necessary statements have been included in the suitable portion of this manuscript. The roles of both GIP drugs and imeglimin have been encompassed in this manuscript.

Round 2

Reviewer 1 Report

No more requests